# Pro- and Antioxidant Effect of Food Items and Matrices during Simulated In Vitro Digestion

**DOI:** 10.3390/foods12081719

**Published:** 2023-04-20

**Authors:** Farhad Vahid, Lisi Wagener, Bernadette Leners, Torsten Bohn

**Affiliations:** Nutrition and Health Research Group, Department of Precision Health, Luxembourg Institute of Health, 1 A-B, L-1445 Strassen, Luxembourg

**Keywords:** reactive oxygen species (ROS), oxidative stress, simulated digestion, INFOGEST model, inflammation

## Abstract

The digestive tract can be considered a bioreactor. High levels of reactive oxygen species (ROS) during digestion may predispose for local and/or systemic oxidative stress and inflammation, e.g., inflammatory bowel diseases. Food items rich in antioxidants may prevent such aggravation. This investigation analyzed pro-and antioxidant patterns of food matrices/items following in vitro digestion. Gastrointestinal digestion reflecting typically consumed quantities was performed on nine food items (orange and tomato juice, soda, coffee, white chocolate, sausage, vitamin C and E, and curcumin) and their combinations (n = 24), using the INFOGEST model. Antioxidant potential was measured by FRAP, DPPH, and ABTS, and pro-oxidant aspects by MDA (malondialdehyde) and peroxide formation. An anti-pro-oxidant score was developed, combining the five assays. Liquid food items showed moderately high antioxidant values, except for coffee and orange juice, which exhibited a high antioxidant potential. Solid matrices, e.g., white chocolate and sausage, showed both high pro-oxidant (up to 22 mg/L MDA) and high antioxidant potential (up to 336 mg/L vitamin C equivalents) at the same time. Individual vitamins (C and E) at physiological levels (achievable from food items) showed a moderate antioxidant potential (<220 mg/L vitamin C equivalents). Overall, both antioxidant and pro-oxidant assays correlated well, with correlation coefficients of up to 0.894. The effects of food combinations were generally additive, i.e., non-synergistic, except for combinations with sausage, where strong quenching effects for MDA were observed, e.g., with orange juice. In conclusion, as especially highlighted by complex matrices demonstrating both pro- and antioxidant potential, only measuring one aspect would result in physiological misinterpretations. Therefore, it is imperative to employ a combination of assays to evaluate both pro- and antioxidant properties of food digesta to ensure physiological relevance.

## 1. Introduction

Many biological processes in the body but also the environment expose our cells to reactive oxygen species (ROS) [1]. ROS are molecules containing a reactive oxygen group, such as hydrogen peroxide, and are often also termed oxygen radicals [1]. To a certain level, ROS is necessary for biological processes such as cell signaling and immune function; in excess, however, it can lead to cellular damage, such as DNA strand breaks, as well as disturbances in the protein and lipid structures, among other biological damages [2]. The mechanism of ROS formation involves the transfer of electrons from a donor molecule to molecular oxygen, leading to the production of superoxide anion (O_2_•^−^) in a process known as electron leakage. Superoxide anions can be further converted to hydrogen peroxide (H_2_O_2_) by the enzyme superoxide dismutase (SOD). H_2_O_2_ can then be converted to hydroxyl radicals (OH•) in the presence of transition metals, such as iron and copper, in a process known as the Fenton reaction [1]. However, when ROS levels become too high and overwhelm the cell’s antioxidant defense mechanisms, they can cause oxidative stress (OS) and damage cellular structures. For instance, ROS can impair mitochondrial function, a critical cellular process responsible for generating energy in the form of ATP. ROS accumulating in the mitochondria can damage the mitochondrial DNA and disrupt the electron transport chain, leading to impaired energy production and cellular death. This process has been implicated in the etiology of a range of diseases, including a number of metabolic disorders and neurodegenerative diseases [3].

Therefore, oxidative stress (OS) can worsen or even lead to the origin of various diseases, including cardiometabolic ones such as diabetes [4] and cardiovascular diseases [5], or gastrointestinal (GI) diseases, also characterized by inflammatory processes. Of note, there is a close relation between OS and inflammation, with one possibly aggravating the other [6,7]. Notably, ROS can activate cellular signaling pathways that contribute to inflammation and cell death. For example, ROS can activate the NF-κB signaling pathway, which is involved in regulating inflammation and immune responses. Excessive activation of this pathway can lead to chronic inflammation and contribute to developing diseases such as rheumatoid arthritis and inflammatory bowel disease [8], among others, as many chronic conditions are characterized by increased systemic levels of inflammation.

Inflammatory conditions in the GI tract are, therefore, often characterized by an increased amount of ROS, further enhanced by the presence of immune cells such as T cells [9]. So far, no cure has been found for GI diseases such as inflammatory bowel disease (IBD), including particularly ulcerative colitis and Crohn’s disease, which eventually may lead to cancer [10]. In line with ROS playing a role in the progression of these diseases, antioxidant therapies have been shown to improve these conditions in part [11]. For example, resveratrol and curcumin have been employed in treating IBD [12]. Indeed, several secondary plant metabolites have been shown to activate Nrf2, a transcription factor that may trigger the body’s own antioxidant defense system, upregulating, among others, antioxidant enzymes such as CAT, SOD, and HO-1 [13]. Therefore, in addition to any possible direct quenching effects that certain antioxidants such as polyphenols and carotenoids may have, such interactions with transcription factors and/or nuclear receptors [14,15] are also paramount.

A number of studies have described harmful pro-oxidant effects of certain food items upon their ingestion [16,17]. For instance, intestinal OS after a high-fat diet was investigated in a mouse model, and the authors found an increase in ROS related to an upregulation of the NADPH oxidase enzyme in the mucosa of the colon [18]. Dietary fats can be a source of peroxides and further products, such as malondialdehyde (MDA), which can cause adduct formation with cellular membranes, proteins, and DNA [19]. Along with fats, sugars have also been linked to ROS-induced OS on a systemic level [20]. In addition, a recent in vitro study highlighted that glucose increased protein oxidation/nitration during simulated gastric digestion, and sugars may aggravate oxidative/nitrosative reactions and disturb the digestion of red meat [21]. It should be noted that even food items with ascribed antioxidant properties, such as vitamins C and E and polyphenols, can also exhibit pro-oxidant properties under certain physiological conditions [22].

Therefore, measuring both antioxidant and pro-oxidant properties of food items can better describe their behavior in the GI tract or even at the systemic level. Many methods are available for determining food antioxidant and/or pro-oxidant capacity; however, combining several in vitro methods measuring different aspects may reflect more accurately physiological properties. While FRAP works by electron exchange [23], ABTS and DPPH rather capture aspects of hydrogen exchange [24]. Thus, no single test would reflect the array of antioxidant or pro-oxidant reactions in vivo. Although several assays have been used in some studies to investigate antioxidant and/or pro-oxidant effects, to the best of our knowledge, no score/index combining both aspects has been developed to date. One study proposed a relative antioxidant capacity index (RACI) to evaluate food antioxidant capacity using seven tests (ABTS, FRAP, TRAP, phenol antioxidant index, ORAC (Cu^2+^), ORAC (ROO^•^), and ORAC (OH^•^)) [25]. However, one of the main limitations of that study was that it was only based on available, published data, assuming that experimental conditions were similar and experimental errors were ignored for the same method, and tests focused only on antioxidant properties. Furthermore, food items were not studied following digestion.

As the digestive tract is the first contact and interface of nutrients with tissues of the human body, it is also here that the concentration of secondary plant metabolites may be the highest, as many of these compounds are only incompletely absorbed or rapidly metabolized or excreted [26,27]. Through digestive processes, a certain level of ROS is produced during digestion, which could, following uptake via the gut, exert adverse systemic effects on health. Interestingly, there is very little information about ROS production and uptake during and following digestive processes. Many studies have merely looked at the antioxidant activity before digestion, though this is subject to strong changes during digestion, such as protein digestion into amino acids [28] or liberation of more strongly bound phenolics [29]. Most studies evaluating OS have looked at blood plasma levels and isolated cell cultures, and there is an evident lack of understanding of how pro-oxidants and antioxidants influence the GI tract prior to absorption and whether they could have an effect at a local level, such as on GI diseases or later, following absorption, at the systemic level.

Therefore, the present study aimed as a first step to investigate different food items and their combinations regarding antioxidant and pro-oxidant aspects during GI digestion. For this purpose, we studied pro-and antioxidant food items and their combinations following their simulated GI digestion by various antioxidant tests and also measures of pro-oxidant potential with respect to lipid oxidation. It was hypothesized that food matrices containing recognized antioxidants would have a higher antioxidant potential that remains following simulated digestion and that matrices containing lipids and those of lower antioxidant potential would show higher concentrations of peroxides and resulting reactive pro-oxidants such as malondialdehyde (MDA).

## 2. Materials and Methods

### 2.1. Chemicals, including Enzymes Used for the Digestions and Analyses

Vitamin E was purchased from Thermo Fisher Scientific (Merelbeke, Belgium) at >97% purity; vitamin C was obtained from Sigma-Aldrich (St. Louis, MO, USA) at 99% purity; curcumin was bought from Apollo Scientific (Bredbury, UK) at 98% purity; porcine bile extract (product no. B8631), pancreatin (4× USP specifications, product no. P1750), and pepsin from porcine gastric mucosa (>250 units/mg activity, product no. P7000) were purchased from Sigma-Aldrich. Unless otherwise stated, all other employed chemicals were from Sigma-Aldrich.

### 2.2. Food Items and Matrices

Food items were chosen based on the frequency of consumption within Westernized diets and usability aspects regarding the experimental set-up and their expected contrasting pro-and antioxidant properties. Toward this end, we used data from two consecutive cross-sectional national surveys that were conducted in 2007–2008 and 2016–2017 that represent food consumption in Luxembourg [30,31]. The results of those studies [30,31] showed that dietary patterns in Luxembourg are becoming more Westernized in line with other European countries, including the intake of larger amounts of processed foods such as sausages, saturated fat, and high-sugar items such as white chocolate or soda. All food items were purchased from a local supermarket (Delhaize, Strassen, Luxembourg) and were used as sold (without further kitchen processes or other preparations). The amounts used for solid and liquid food items were based on the standard serving size of that food divided by 40 (the ratio of 1 L of assumed GI fluids by the amount used in our digestion experiments, i.e., 25 mL). For vitamins C and E, the highest amount that could be obtained from a single food item (orange juice and walnut, respectively), and for curcumin, intake from a supplement of 500 mg, were considered. Selected food items, their quantity, and combinations are shown in Table 1.

Each food item (n = 9) was tested individually, and food items related to the Western diet, including sausage, soda, and white chocolate (SSW) (n = 3), were tested in combination with each antioxidant (n = 6), leading to 27 different conditions (9 alone and 18 in combination). Six additional conditions were tested with either two items of the SSW group or two antioxidants. Ultimately, 33 analyses were performed on 9 individual food items or their combinations (n = 24). Food combinations and further details of used products are listed in Appendix A.

### 2.3. In Vitro Digestion

The entire procedure followed the INFOGEST consensus model for static digestion [32]. The digestion protocol is described briefly in the following.

#### 2.3.1. Stock Solutions and Simulated Digestive Fluids

Seven stock solutions to create the simulated digestive fluids were prepared with a volume of 100 mL each, except for NaHCO_3_, of which 250 mL were prepared, according to the concentrations and quantities described in Appendix A. They were added to volumetric flasks, filled up to around 80 mL with MilliQ water (Merck, Darmstadt, Germany), and put on a magnetic stirrer to dissolve. With a 100 mL volumetric flask and MilliQ water, the volume was adjusted to 100 mL. Simulated gastric fluid (SGF) and simulated intestinal fluid (SIF), consisting of varying physiological minerals and salts, were prepared fresh each day with the stock solutions. CaCl_2_(H_2_O)_2_ was not added to the digestive solutions, as it was added separately to the digestive conditions in a later step to prevent precipitations.

#### 2.3.2. Simulated Gastrointestinal Digestion

The digestions were performed in 50 mL tubes (Falcon, Greiner Bio-One, Vilvorde, Belgium). Typically, 20 tubes were treated in parallel. Each condition was performed in quadruples, meaning 5 conditions could be tested. On each day of experiments, blanks (no food matrix added) and controls (2 mL tomato juice + 2 mL of MilliQ water) were run, plus 3 conditions of interest. Enzyme solutions were prepared by dissolving 2180 mg of porcine pepsin in 21.8 mL of SGF for the gastric phase. For the intestinal phase, 1060 mg of pancreatin, along with 3646 mg of porcine bile, was dissolved in 53 mL of SIF. Solutions were put on a magnetic stirrer until the powder was dissolved.

#### 2.3.3. Gastric Phase of Digestion

In the case of liquid samples, 2 mL were used; for solid samples, 2 g were used, and with vitamins, C, E, and curcumin where only a very small amount (the highest amount that can be obtained from a single food item) was used, another 2 mL of MilliQ was added. This led to starting conditions of 4 mL volumes. If foods were analyzed by themselves (individually) and not in combination, another 2 mL of MilliQ was added to reach 4 mL. All digestions were performed in 50 mL falcon tubes. In addition, 1.04 mL of pepsin solution was added to each tube. Subsequently, 4.16 mL of SGF solution and 33 μL of CaCl_2_ (0.03 M) were added. Samples were vortexed, and the pH was adjusted to a value of 3 ± 0.4 with HCl (1 M). The volume for gastric digestion was adjusted to 13 mL with MilliQ. For incubation of the gastric phase, the samples were submerged for 2 h at 37 °C in a horizontal position in the direction of the movement in a shaking water bath (GFL, Burgwedel, Germany) at 100 rpm.

An exception to this procedure was vitamin E and its combinations. The chosen 1.5 mg was too low to weigh reliably as it was challenging to pipet due to its viscous nature. Due to this, the gastric phase for vitamin E was prepared only in one tube with 6 mg of vitamin E along with 10 mg of Tween20 to foster emulsification. In addition, 4.16 mL of pepsin, 16.6 mL of SGF, and 130 μL of CaCl_2_ were added. Compared to conditions prepared in quadruple immediately, vitamin E conditions were not filled up with MilliQ before incubation but at the beginning of the intestinal phase. After 1 h, the pH of the digesta was checked to ensure it remained stable. If there was a variation greater than 0.4 units, it was adjusted again by 0.1 mol/L NaOH or HCl.

#### 2.3.4. Intestinal Phase of Digestion

Vitamin E conditions previously prepared only in one tube were split up evenly into 4 tubes and filled up to 13 mL with MilliQ. Then, 2.6 mL of the pancreatin/bile solution was added to each tube with 7.8 mL of SIF and 260 μL of CaCl_2_ (0.03 M). pH was measured and adjusted to a value of 7 ± 0.4 by adding NaOH (1 M) or HCl (1 M), and the final volume was adjusted to 26 mL with MilliQ water. Samples were again incubated with the exact instructions as in the previous step for 2 h, checking and re-adjusting (where needed) the pH after 1 h. Following digestion, aliquots of the entire digestion were immediately frozen at −80 °C for sample storage until further analyses.

### 2.4. Antioxidant Capacity Measurements

#### 2.4.1. FRAP

The FRAP assay utilizes the ability of an antioxidant compound to reduce ferric iron (Fe^3+^) to ferrous iron (Fe^2+^) as a basis for measuring the antioxidant potential in samples. To maintain iron solubility, the FRAP reaction is conducted under acidic conditions at a pH of 3.6. This low pH environment reduces the ionization potential that drives hydrogen atom transfer and enhances the redox potential, which is the main reaction mechanism [33].

The method described by Bouayed et al. was used [23]. In short, an acetate buffer (300 mmol/L, pH 3.6) was prepared by weighing 163.25 mg of sodium acetate trihydrate and dissolving it with 1430 μL of 96% acetic acid (Merck) in 80 mL of MilliQ water. pH was measured and adjusted to 3.6 with NaOH (1 M), and the volume was brought to 250 mL with MilliQ water. A mix of 2,4,6-tri(2-pyridyl)-1,3,5-triazine (TPTZ) and HCl was prepared by dissolving 7.8 mg of 19.5 mg TPTZ in 6.25 mL of 40 mM HCl (VWR Chemicals, Leuven, Belgium). The iron (III) chloride solution was prepared by weighing 33.5 mg of iron (III) chloride hexahydrate and dissolving it in 6.25 mL of MilliQ water.

Iron (II) chloride stock solution was prepared by weighing 12.7 mg of iron (II) chloride and dissolving it in 5 mL of water. The stock solution was used to create 9 standards according to the quantities indicated in Appendix A.

Samples were thawed and sonicated for 10 min in room temperature water and vortexed. The FRAP reagent was prepared by adding 62.5 mL of acetate buffer and 6.25 mL of iron (III) chloride hexahydrate to the 6.25 mL TPTZ-HCl mix. The reagent was heated to 37 °C for 15 min in a water bath (GFL, VWR), during which it was shaken by hand every couple of minutes. It was then left to cool down for ±5 min on ice. Once prepared, it had to be used within 60 min.

Eppendorf tubes, one for each standard/sample, were filled with 750 μL of FRAP reagent, and the standards/samples were added to their respective tubes (100 μL); the mixture was vortexed and incubated at room temperature for 30 min. Following incubation, the tubes were centrifuged (2000× *g*, 3 min) at room temperature, and 300 μL of each tube was put into a well of a transparent 96-well plate (Greiner Bio-One). With the SpectraMax M2 microplate reader (Molecular Devices, San José, CA, USA), absorbance was measured for each well at 605 and at 800 nm; results were expressed as equivalents of iron (II)chloride.

#### 2.4.2. ABTS

The ABTS radical is produced through a reaction between the ABTS salt and a potent oxidizing agent such as potassium persulfate or potassium permanganate. The degree to which the characteristic long-wave absorption spectrum of the blue-green ABTS radical is diminished by antioxidants that donate hydrogen is used to quantify its reduction [34].

ABTS was performed as described by Bouayed et al. [23]. Briefly, the ascorbic acid stock solution of 10 mg/mL was prepared by weighing 100 mg of L-ascorbic acid and dissolving it in 10 mL of MilliQ. Standards were prepared with the data from Appendix A.

The radical solution was created in a 100 mL glass bottle by dissolving 20 mg of 2,2′-azobis(2-amidinopropane) dihydrochloride (AAPH) and 100 mg of 2,2′-azino-bis(3-ethylbenzothiazoline-6-sulfonic acid (ABTS) in 100 mL of DPBS (Lonza, Basel, Switzerland). The solution was heated to 73 °C in a water bath for 35 min while making sure to shake it in 5 min intervals. After cooling for 15 min at room temperature, the solution was split into two 50 mL falcon tubes and centrifuged for 4 min at room temperature (2000× *g*). The supernatant was recovered, and the tubes were then covered in aluminum. One well of a transparent 96-well plate was filled with 300 μL of DPBS, and two others were filled with 300 μL of a radical solution. Absorbance was measured at 750 nm to ensure absorbance was at 1.0 ± 0.2 with the POLARstar OPTIMA (BMG LabTech, Ortenberg, Germany) microplate reader; results were expressed as equivalents of ascorbic acid. Then, 980 μL of the radical solution was pipetted into Eppendorf tubes. Next, 20 μL of MilliQ and 20 μL of the standard solutions or the samples were added, respectively. The tubes were vortexed and incubated for 15 min in a shaking water bath (GFL) (37 °C, 100 rpm).

Tubes were centrifuged for 3 min at room temperature (8000× *g*). The clear 96-well plates were filled with 3 blanks (300 μL) and 300 μL of each standard and sample. Absorbance was measured at 740 nm and also at 485/920 nm (to account for potential turbidity of samples) with a microplate reader (POLARstar OPTIMA).

#### 2.4.3. Peroxide Formation

The peroxides assay measures the concentration of peroxides in a sample and is commonly used to assess the level of oxidation in oils, fats, and other food products. Peroxides are formed due to the reaction between unsaturated fatty acids and oxygen. Antioxidants can slow down or prevent this process by reacting with free radicals that are produced during the oxidation process [35].

The peroxide test was performed with the “Red Hydrogen Peroxide Assay Kit” from Enzo Lifesciences (Zandhoven, Belgium). The reagents were prepared following the kit’s instructions. The substrate stock solution was prepared by adding 250 µL of DMSO into 1 vial of red substrate; the substrate was then aliquoted, and the aliquots were frozen at −20 °C. The peroxidase stock solution was prepared by adding 1 mL of assay buffer into the vial of horseradish peroxidase. The solution was aliquoted, and the aliquots were frozen at −20 °C. For the standards, a stock solution of 20 mM H_2_O_2_ was prepared by diluting a 3% solution given by the kit. A standard curve was prepared with this stock solution to obtain 10, 3.33, 1, 0.33, and 0.1 µM, respectively. The H_2_O_2_ reaction mix needed for a 96-well plate was prepared by mixing 50 µL red peroxidase substrate stock solution, 200 µL peroxidase stock solution, and 4.75 mL assay buffer.

For the reaction, 50 µL of standard or sample was added in the wells, then 50 µL reaction mixture. The plate was then incubated for 30 min at RT and protected from light, and the fluorescence was measured at 540 nm (excitation) and 590 nm (emission) with the SpectraMax M2 microplate reader; units were expressed as equivalents of hydrogen peroxide.

#### 2.4.4. MDA

The mechanism involves a process whereby unsaturated lipids are oxidized to form additional radical species as well as toxic by-products that can be harmful to the host. The MDA (malondialdehyde) assay is a widely used method for measuring lipid peroxidation, which is a process that involves the degradation of polyunsaturated fatty acids in biological membranes by ROS. MDA is one of the end products of lipid peroxidation and is a highly reactive compound that can react with proteins, nucleic acids, and other molecules. The MDA assay is based on the reaction of MDA with thiobarbituric acid (TBA) to form a pink-colored chromophore that can be quantified spectrophotometrically. The reaction between MDA and TBA occurs under acidic conditions, and the resulting chromophore has a maximum absorbance at 532 nm [36].

All reagents and chemicals for the TBARS assay kit (TCA method) were purchased in a kit from Cayman Chemical (Sanbio, Uden, The Netherlands). Ten mL of TBA acetic acid were pipetted into a 50 mL falcon tube along with 40 mL of ultra-pure water for dilution. For the sodium hydroxide assay reagent, 10 mL of NaOH reagent (3.5 M) was diluted with 40 mL of MilliQ water, in which the water was added first and then the NaOH.

The color reagent was prepared by weighing 152.4 g of thiobarbituric acid (TBA), which was added to 14.38 mL of the diluted TBA acetic acid solution in a 50 mL falcon tube (these quantities depended on the number of wells we used in the experiment. These were the amounts needed for 96-wells). The tube was put on a shaker (DLAB Scientific, Beijing, China) to mix until dissolution. A stock solution of 25 μM was created by diluting the 50 μL of MDA standard with 950 μL of MilliQ water. MDA reagent tubes were created according to the concentrations indicated in Appendix A. Then, 50 μL of sample or standard, respectively, was put into 1.5 mL Eppendorf tubes (Hamburg, Germany). In addition, 50 μL of TCA reagent was added to each tube, and the tubes were vortexed. Another 400 μL of color reagent was added per tube. The tubes were closed, vortexed, and left to incubate for 1 h at 99 °C in a thermomixer (Eppendorf) and then put on ice for 10 min. For 4 min, the vials were centrifuged at 5000× *g* and 4 °C. Then, 300 μL of the supernatant was transferred into the wells of a 96-well black plate (Greiner Bio-One), and fluorescence was determined at an excitation wavelength of 525 nm and an emission wavelength of 565 nm, with auto-cut off, with a microplate reader (Molecular Devices).

#### 2.4.5. DPPH

The assay is based on the measurement of the scavenging capacity of antioxidants towards it. The odd electron of the nitrogen atom in DPPH is reduced by receiving a hydrogen atom from antioxidants to the corresponding hydrazine [37].

For the DPPH test, we used the “DPPH Antioxidant assay kit” from Dojindo Molecular Technologies Inc. (Rockville, MD, USA). The stock solutions were prepared following the kit’s instructions, but the protocol was slightly changed to follow our needs. The DPPH working solution was prepared by dissolving and sonicating the DPPH reagent in 10 mL ethanol. The Trolox standard solution was also dissolved in 10 mL ethanol, and a standard curve with the following concentrations was prepared by diluting with ethanol: 100, 75, 50, 25, 10, 5, and 1 µg/mL. For the assay, we added 20 µL of standard or sample into the wells, then added 80 µL of assay buffer and 100 µL of DPPH working solution. The plate was incubated for 10 min on a shaking thermomixer (500 rpm), and the absorbance was measured with the POLARstar OPTIMA at 520 and 750 nm (for the subtraction of potential turbidity); units were expressed as Trolox equivalents.

#### 2.4.6. Anti-Pro-Oxidant Score

To assess the balance between pro-oxidants and antioxidants following gastrointestinal digestion, we developed a score based on the Z-Score of all 5 measurements. First, we obtained the Z-score of the 5 measurements. For equal weighting for pro- and antioxidant measurements, we multiplied each equation by 1/2 and then divided each equation by the number of its components. Therefore, we used the following formula to calculate the score:$$\mathrm{Anti-pro-oxidant score}=\frac{0.5\times \left(ZABTS+ZFRAP+ZDPPH\right)}{3}-\frac{0.5\times \left(ZPeroxides+ZMDA\right)}{2}$$

### 2.5. Data Treatment and Statistical Analysis

Raw values were corrected by subtracting the respective blank values. A Grubbs’ test was performed to eliminate any outliers. A normalization and adjustment procedure was carried out to remove day-to-day variation from the investigation. For this purpose, the individual concentrations obtained were divided by the mean of the daily control (tomato juice) and multiplied by 100. Then, to adjust values back to realistic concentrations, these values were divided by 100 and multiplied by the global mean concentration found for the control samples (tomato juice).

Normality of distribution and equality were tested by Q-Q plots and box-plots, respectively. Linear multivariate models were created to study the effect of food items and their combinations on FRAP, ABTS, DPPH, peroxides, and MDA. These models included the sample type (e.g., orange juice, etc.) as a fixed factor and FRAP, ABTS, DPPH, peroxide number, and MDA as observed dependent factors. Furthermore, pairwise comparisons were carried out between the laboratory digestion combinations and the mathematical combinations of the individually digested food items to detect additive or synergistic effects by employing Student’s unpaired *t*-tests. Tukey’s post hoc tests were carried out following significant Fisher-F tests to allow for studying all group-wise comparisons. Data were analyzed with SPSS (IBM, Chicago, IL, USA) version 25.0. A *p*-value of <0.05 (2-sided) was considered statistically significant.

## 3. Results

### 3.1. Data Distribution and Results from the Overall Multivariate Test

Based on Q-Q and box plots, distributions were not normal, and data were log-transformed. When testing the significance of conditions for all five measurements (and developed score), a significant *p*-value (*p* < 0.001) was found, indicating that the outcomes were significantly different depending on the digested test meals and their combinations. The correlation matrix between the pro- and antioxidant measurements (all five measurements and developed score) is shown in Figure 1.

### 3.2. Findings from Assays

#### 3.2.1. FRAP

Based on the multivariate analysis, FRAP was significantly influenced by the test meal (*p* < 0.001). Following post hoc test analyses (Tukey’s), a number of homogeneous subsets were found, as indicated in Figure 2. Solid matrices generally exhibited higher FRAP values than liquid matrices, which in turn had higher values than vitamins added at physiological levels, with an overall range varying from 39.9 mg/mL Fe^2+^ (blank) to 182.1 mg/mL (coffee). Soda showed the lowest (41.8 mg/mL) antioxidant potential of the studied food items. Interestingly, the blank also showed an antioxidant effect by itself.

Regarding laboratory combinations, sausage plus coffee showed the highest antioxidant potential, up to 311.7 mg/mL of Fe^2+^, followed by the combination of coffee and white chocolate (306.8 mg/mL). The lowest values for combined food items were found for soda plus vitamin E (45.3 mg/mL) (Figure 3).

Studying additive effects, it was also apparent that FRAP values in combinations of two food items or constituents were not fully additive but generally summed up to slightly less than the sum of the original two FRAP values (Figure 4). White chocolate’s antioxidant capacity interestingly appeared to decline with the addition of vitamin E (*p* < 0.001). Overall, values for mathematical combinations (blank subtracted) ranged from soda and curcumin (10.1 mg/mL) to coffee and orange juice (216.3 mg/mL).

#### 3.2.2. ABTS

The ABTS assay results mirrored (the highest correlation was between FRAP and ABTS (*ρ* = 0.901)) the FRAP assay findings, with the highest values for solid foods, lower values for liquid food items, and even lower ones for added vitamins, ranging from 202.4 (blank) to 296.2 mg/mL (white chocolate) (Figure 2).

Regarding laboratory combinations, vitamin E plus soda displayed the lowest antioxidant potential (192.2 mg/L), only showing non-significant differences to the blank. Coffee plus white chocolate showed the highest amount of vitamin C equivalents (VCE) with a mean of 333.8 mg/L (Figure 3). Along with solid matrices, the liquid combination of coffee with orange juice also showed a high level of antioxidants with 327.0 mg/L (Figure 3).

Studying additive effects, as for FRAP, adding vitamin E to white chocolate had no further additional positive effects on antioxidant activity (Figure 4). As for FRAP, the effects of matrices were not fully additive (Figure 4).

#### 3.2.3. DPPH

Concerning DPPH, individual results with a minimum of 27.7 mg/mL (for blanks) to a maximum of 96.6 mg/mL (for coffee) were very similar in tendency to the FRAP and ABTS findings; perhaps the only marked difference was curcumin, which showed a high antioxidant effect (Figure 2).

Regarding laboratory combinations, the results ranged from 16.9 (for white chocolate plus orange juice) to 101.3 mg/mL (for coffee plus white chocolate) (Figure 3).

Studying additive effects, the mathematical combination of soda and vitamin C showed the lowest (11.1 mg/mL), and the combination of sausage and curcumin (134.1 mg/mL) showed the highest (blank corrected) values (Figure 4).

**Figure 2 foods-12-01719-f002:**
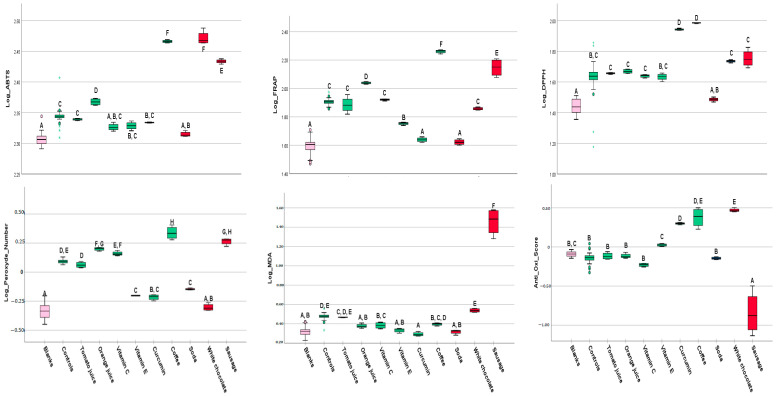
Multivariate analysis comparing antioxidant and pro-oxidant capacity of individual food items after in vitro digestion. Boxes not sharing the same superscript (capital letters) are statistically significantly different (Tukey’s post hoc test, *p* < 0.05). Log = All values are in logarithmic form. Pink color indicates the respective blank. Green colors represent presumably antioxidant-rich food items, red ones presumably pro-oxidant items.

#### 3.2.4. MDA

Again interestingly, empty digesta did not show zero concentrations, indicating the production of MDA based on “empty stomach” conditions (Figure 2). Results were similar to ABTS and FRAP in that the highest concentrations were found for sausage (29.7 mg/mL) and the lowest concentration for blanks (2.1 mg/mL) (Figure 2).

Regarding laboratory combinations, sausage and chocolate, and their combinations resulted in the highest MDA concentrations (up to 37.5 mg/L for vitamin C plus sausage), possibly in line with their higher lipid levels in the matrix (Figure 3). As in the other two tests, solid matrixes exhibited the highest values, while vitamins along with soda were on the lower spectrum of measured MDA levels, with a minimum of 1.9 mg/mL for the combination of curcumin with soda (Figure 3).

Studying additive effects, the mathematical combination of curcumin with soda showed the lowest (around zero mg/L), and sausage + white chocolate (28.3 mg/L) showed the highest (blank corrected) MDA values (Figure 4). In combinations, MDA production in soda was significantly elevated when combined with sausage. Interestingly, MDA levels in sausage were significantly lowered when digested with other rather antioxidant-containing matrices, including vitamin E, orange or tomato juice, curcumin, coffee, and surprisingly, also white chocolate (Figure 4).

#### 3.2.5. Peroxide Assay

Results regarding the peroxide concentration were in line with the results of MDA, and values ranged from a minimum of 0.46 mg/mL (for blanks) to a maximum of 2.1 mg/mL (for coffee) (Figure 2).

Laboratory combinations showed that vitamin E + soda had the lowest (0.46 mg/mL) and vitamin C + sausage (3.0 mg/mL) had the highest values (Figure 3).

Regarding additive effects, the mathematical combination of curcumin + white chocolate (0.07 mg/mL) showed the lowest, and coffee + orange juice (2.74 mg/mL) showed the highest (blank corrected) values for peroxides (Figure 4).

**Figure 3 foods-12-01719-f003:**
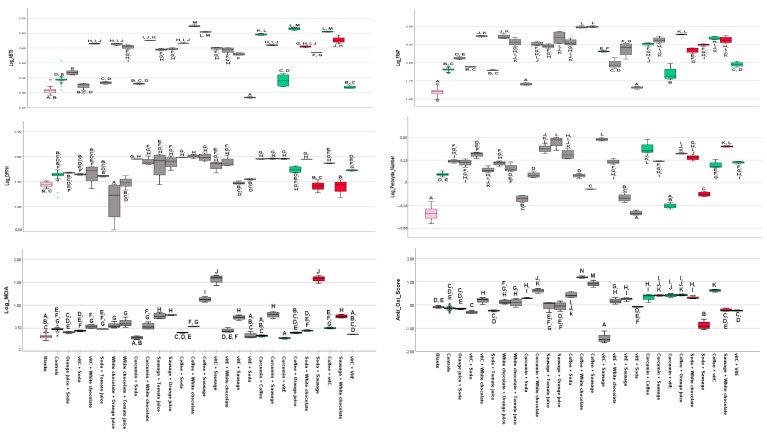
Multivariate analysis comparing antioxidant and pro-oxidant capacity of laboratory combinations of food items after in vitro digestion. Boxes not sharing the same superscript (capital letters) differ statistically significantly (Tukey’s post hoc test, *p* < 0.05). Log = All values are in logarithmic form. Pink color indicates the respective blank. Green colors represent presumably antioxidant-rich food combinations, red ones presumably pro-oxidant food combinations, and grey colors indicating a combination of one pro-and one antioxidant food item.

**Figure 4 foods-12-01719-f004:**
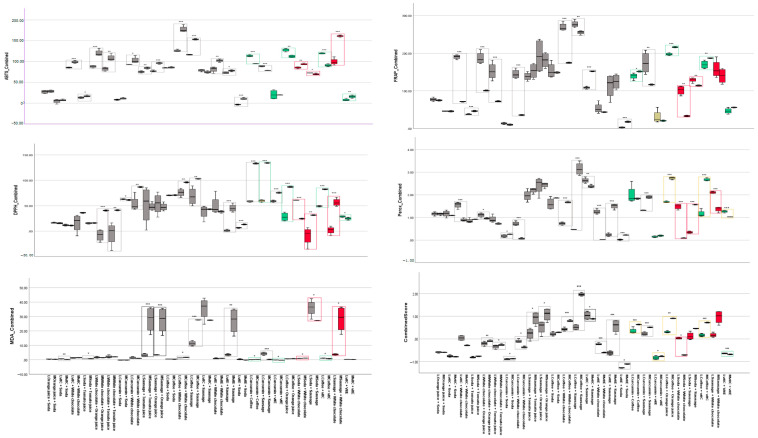
Pairwise (laboratory combinations vs. mathematical combinations) comparisons using *t*-tests. Note that all values were baseline (control digestion) subtracted. *p*-values: * < 0.01, ** < 0.001, *** < 0.0001. L = laboratory combinations, i.e., digested together, M = mathematical combinations. Pink color indicates the respective blank. Green colors represent presumably antioxidant-rich food combinations, red ones presumably pro-oxidant food combinations, grey colors indicate a combination of one pro-and one antioxidant food item.

## 4. Discussion

In the present study, the digestion of several frequently consumed food items and their combinations was simulated by means of an accepted in vitro model, simulating the gastric and intestinal phases (INFOGEST consensus model). In the following, the antioxidant capacity after digestion was measured for food items and their combinations with the FRAP, ABTS, and DPPH assays, and pro-oxidant measures regarding lipid oxidation/peroxidation were carried out by the MDA-TBARS and the peroxide assay.

The most important finding was that some food items, including sausage, white chocolate, and also coffee, showed both among the highest pro-oxidant and antioxidant values at the same time. This is an important strength of this study in that both antioxidant and pro-oxidant measures were carried out, which is often neglected [38,39]. Contrarily, liquid matrices showed a relatively low antioxidant capacity, while added vitamins and curcuminoids, in the concentrations expected to occur in the food matrices studied (e.g., orange juice), showed rather low antioxidant and low pro-oxidant patterns. When comparing combinations of digested items under laboratory conditions, it was found that combinations of sausage or white chocolate with coffee had the highest, while sausage with vitamin C and sausage with soda had the lowest anti-pro-oxidant score. Similar results were obtained for the mathematical combinations. Comparing the laboratory combinations with the mathematical combinations showed that, in almost all cases, rather additive, i.e., non-synergistic effects were present.

To the best of our knowledge, very limited studies, if any, have systematically investigated the effects of oxidative and/or antioxidant effects of food items and matrices during digestion, using both several pro-and antioxidant endpoints. Since the digestive system is the first place that food comes into contact with the human body, the effects of pro-oxidant and/or antioxidant activity of food items are the first location where detrimental effects may originate that later could have adverse effects on cellular systems. Thus, synergistic pro- or antioxidant effects of food constituents in the digestive system may be an important pre-determining factor that can exert systemic effects on the body [40,41].

A recent study determined the effects of antagonism, synergism, and additive interactions between food items by statistically comparing estimated and measured total antioxidant capacity (TAC) by the ABTS assay [42]. In pre-digestion estimates, the authors reported an antagonism in the combinations of milk with green tea extract or fruits, and an obvious synergistic effect was observed for the combination of fruits with whole wheat bread, breakfast cereal, or yogurt [42]. In the digested experiments, however, they reported that somewhat alkaline conditions significantly increased the TAC of food items. The authors highlighted that, e.g., protein–phenol interactions (polyphenol–protein complex formation) provided an antagonism of milk with fruits or green tea extract prior to digestion, but a synergism in milk combinations (with fruits, breakfast cereal, whole wheat bread, or yogurt following digestion [42], emphasizing changes occurring during digestion.

Interestingly, the digestion of the blank samples without food constituents already showed a certain antioxidant as well as pro-oxidant activity in our study. While there is a lack of research on the pro-oxidant effects of bile and digestive enzymes, several studies have found bile components, such as bilirubin, to have an antioxidant effect due to their ability to bind free radicals, e.g., reactive nitrogen species (RNS) [43,44]. For example, a recent study investigated the in vitro antioxidant effects of two different types of synthetic bilirubin using FRAP and ABTS assays [44]. The authors reported that both compounds exhibited similar potent antioxidant effects [44]. Bilirubin can be oxidized to biliverdin, which is immediately reduced by biliverdin reductase to bilirubin in biological systems [45]. Another study concluded that bilirubin primarily protects against lipid peroxidation [45]; however, it is still a likely antioxidant, which may explain our findings of the empty digesta. Besides this protective effect, bilirubin has also been found to act as a pro-oxidant, causing DNA strand damage [46].

Indeed, a critical and debatable issue about the pro-oxidant and antioxidant properties of food in the digestive tract is that, although the antioxidant activity of food components, e.g., phytochemicals, is well recognized, they can also display pro-oxidant activities under certain conditions, such as at high doses or in the presence of metal ions [47]. In addition, in vitro, it has been shown that the pH influences oxido-reductions, suggesting that the pH of biological tissues could impact the antioxidant/pro-oxidant and chelating activities. For example, a decrease in pH causes a reduced chelating effect of phenolics toward iron, possibly due to increased solubility of the complexes and replacement of the iron by protons [47].

Regarding the high antioxidant and pro-oxidant values for sausage in our study, an earlier study showed that heat treatment during sausage processing could significantly increase measured markers of oxidation [48]. Processing operations and lipid formulation could significantly impact in vitro digestibility and chemical and functional properties within the complex interrelationships between oxidation mechanisms. For instance, stabilizers such as nitrite in processed meats may inhibit lipid oxidation; e.g., in a cooked muscle system, 50 mg/kg nitrite resulted in a significant reduction in MDA concentrations [49]. Similar and comparable antioxidant effects of nitrite were observed in heated water-extracted pork muscle systems. Nitrite appears to ‘chelate’ non-heme iron forming a stable complex, thus inhibiting catalytic activity [49]. The potential effect of nitrite curing of meat on lipid oxidation during gastrointestinal digestion was investigated earlier [50], showing that nitrite curing of pork, chicken, and beef (high sources of heme-Fe) inhibited oxidation during digestion. The authors further confirmed the catalyzing effect of heme-Fe in meat on oxidative stress and demonstrated NOC-induced DNA damage during digestion. Nitrite curing of meat resulted in a lower formation of toxic oxidation products [50]. The antioxidant mechanism of nitrite is based on the antioxidant activity of the formed nitric oxide myoglobin complex, nitric oxide ferrous complexes, and S-nitrosocystein and inhibition of the Fenton reaction, which is responsible for the initiation of oxidation reactions [51]. Furthermore, a stabilizing effect of nitrite was observed on the susceptibility of unsaturated lipids in the membranes to oxidation [52]. In acidic conditions, such as those present in the stomach, nitrous acid generates dinitrogen trioxide (N_2_O_3_) and H_2_O, which is in equilibrium with nitric oxide (•NO) and nitrogen dioxide (•NO_2_). A dual role of •NO on lipid oxidation was described whereby a 1:1 ratio of •NO to ROS enhances lipid peroxidation, whereas an excess of •NO inhibits oxidation [52]. Finally, in addition to nitrite, ascorbic acid is added during meat processing, including sausage, to ensure the nitrates’ red coloring effect, but also antioxidant properties are enhanced [53]. Therefore, the high antioxidant activity of sausage in the present experiments is likely attributed to the content of nitrite, ascorbic acid, and also the high protein content that could contribute following digestion and hydrolysis to antioxidant properties [54].

White chocolate, like sausage, contains high amounts of saturated fatty acids and a fair number of proteins, and a similar interpretation as for sausage can be imagined for its pro-and antioxidant mechanisms, e.g., production of end products of lipid peroxidation, such as 4-hydroxynonenal and MDA. MDA, at its physiological state and neutral pH, appears as an enolate anion and shows low chemical reactivity. However, following absorption by tissues, MDA interacts with nucleic acid bases and can form several different adducts, which may induce sequence-dependent frameshift mutations and base-pair substitutions in mammalian cells [55]. An alternative mechanism of genotoxicity is related to the ability of MDA to create interstrand cross-links in DNA [55]. Accumulating evidence suggests that MDA-altered epitopes are pro-inflammatory and thus important targets of adaptive and innate immune responses, therefore serving as interesting potential targets for immunological therapeutic interventions in CVD [56].

For chocolate and sausage, during digestion, the presence of proteins in the food bolus could significantly influence the extent of lipid oxidation, increasing or decreasing it depending on the protein’s nature [57]. For instance, proteins containing sulfur or aromatic groups may inhibit lipid oxidation and can do this through multiple pathways, including the inactivation of ROS, scavenging free radicals, chelation of pro-oxidative transition metals such as iron, and reduction of hydroperoxides [58]. Again, the production of peptides through hydrolytic reactions seems to be the most promising technique to form proteinaceous antioxidants, since peptides can have substantially higher antioxidant activity than intact proteins [58]. The high MDA values of the white chocolate may be explained by the presence of at least 20% cocoa butter, presenting both monounsaturated and saturated fats [59]. Additionally, white chocolate is also high in sugars, which could also lead to increased antioxidant potential in assays such as FRAP, which reacts with reducing sugars [60].

In this respect, most of the presently employed antioxidant tests show a fairly broad reactivity which makes them appropriate to study complex mixtures of food items and their digesta. In addition to FRAP reacting with reducing sugars, it has been reported to react with a large number of food constituents, such as other metals, phenolics, antioxidants such as vitamin C and E, glutathione, and proteins such as albumin, and is employed also to measure TAC [61]. The ABTS assay has been mostly applied to phenolic compounds as it correlates well with the number of hydroxyl groups [62] but has been mentioned to react with other compounds, such as Maillard products [63] or aminothiols [64], similar as for the DPPH assay. With respect to the pro-oxidant tests used, compounds such as sugars, amino acids, oxidized proteins, and esters have been shown to exert an influence on the TBARS assay by interacting with TBA [65]. By comparison, the peroxide assay used measures hydroperoxides such as H_2_O_2_ resulting, e.g., from lipid oxidation and is a relatively specific assay when using enzymes such as HRP as in the present investigation, though higher concentrations of thiols such as present glutathione may be interfering. Regarding MDA, certain compounds could interfere with the reaction between TBA and MDA and/or hinder a proper spectrophotometric measurement, i.e., interferences with water-soluble proteins, peptides, other aldehydes different from MDA, pigments, amino acids, additives, and fat droplets may occur [66].

The low reducing potential for soda was expected, as a soda without added antioxidants (e.g., vitamin C or beta-carotene) high in free sugars was chosen. A review article highlighted that sugar could increase the production of ROS by several mechanisms (through various ways including mitochondria, sorbitol pathway, nicotinamide adenine dinucleotide phosphate (NADPH)-oxidase, activated glycation, and insulin pathway) in vivo, and as a result, it can have pro-oxidant effects [67]. Therefore, cellular models may be more accurate in capturing such pro-oxidant aspects. Furthermore, advanced glycation end products (AGEs) could be formed from the non-enzymatic reaction of reducing sugars with the amino group of proteins, lipids, and nucleic acids [67]. This may occur when combining soda with sausage, white chocolate, or even with digestive enzymes, and can modify their oxidizing properties. For instance, in Maillard reactions, glucose attaches to free amino acids of proteins non-enzymatically to form a Schiff-base that spontaneously rearranges irreversibly to form Amadori products. Some AGE products include Nε-carboxymethyl-lysine, pentosidine, pyrilline, Nε-carboxyethyl-lysine, and argpyramidine [67]. Although there is no evidence that these reactions take place during in vitro digestion, from a mechanistic point of view, such reactions are possible. Future studies should investigate AGE products and their pro-and antioxidant effects.

It was shown that vegetables potentially reduce oxidative phenomena during digestion when combined with meat [68]. In vitro co-digestion of turkey meat with a diet containing tomato, onion, black olives, extra-virgin olive oil, and basil dose-dependently reduced lipid peroxidation. It was concluded that flavonols and anthocyanins were the most effective compounds in reducing pro-oxidant phenomena after co-digestion with meat, showing high hydroxyl radical scavenging activity. By comparison, phenolic acids exhibited the highest ability to reduce Fe^3+^ to Fe^2+^ and showed the lowest lipid peroxidation inhibitory effect [68]. In addition, the amount of fat in meats can affect oxidation during digestion. A study showed that high-fat vs. low-fat beef digests contained approximately 10-fold higher lipid oxidation product concentrations [69]. During digestion of high-fat meat, phenolic acids, including gallic, ferulic, chlorogenic, and caffeic acid, displayed either antioxidant or pro-oxidant demeanor at higher and lower doses, respectively, whereas ascorbic acid acted as pro-oxidant at all doses and the lipophilic reducing compounds, including α-tocopherol, quercetin, and silibinin, all exerted an antioxidant effect. In our experiment, for both FRAP and ABTS, adding coffee to a “pro-oxidant food item” (sausage, soda, white chocolate) resulted in very high antioxidant values and significantly improved antioxidant capacity for all three tested pro-oxidants. Coffee also significantly reduced lipid peroxidation when combined with sausage, in line with literature supporting the antioxidant effects of coffee [70] related to its high amount of phenolic compounds, especially chlorogenic acid and its derivatives, including caffeoylquinic acids. The reduced MDA levels of sausage when combined with coffee, may be explained, in addition to direct peroxide quenching effects, also by inhibiting pancreatic lipase and liberation of free fatty acids [71], though this was not investigated in the present study.

In a similar study, methods using ABTS and the oxygen radical absorbance capacity test (ORAC) measured the capacity of coffee polyphenols to scavenge free radicals [72]. Different caffeine metabolites, such as dihydroferulic acid and m-coumaric acid, showed high antioxidant activity [72]. Another study showed a high and significant correlation between the estimated polyphenol contents of coffee and the ABTS or FRAP values [73]. Due to the heat process during coffee roasting, increasingly significant antioxidant activity was reported due to formed Maillard products [73]. Another study showed that ethanol extracts of coffee had the potential to be used as a natural antioxidant in meat [74]. Similarly, the results of our study showed that the combination of coffee and sausage moderated the pro-oxidant effects of sausage.

The FRAP assay also pointed out that orange juice significantly improved antioxidant levels in all three pro-oxidant food items, as well as in soda, in the ABTS assay. MDA was also significantly lowered in the combination of orange juice with sausage. Orange juice contains not only vitamin C (which exhibited a limited antioxidant effect by itself) but also antioxidants from the carotenoid and flavonoid groups [75]. These findings are in line with an in vivo study that showed significantly elevated antioxidant levels and lowered lipid peroxidation levels in the blood of subjects after orange juice consumption [76].

Both added vitamin E and curcumin did not result in high antioxidant findings. This could be attributed to the low though physiological dose that was chosen. Simons et al. proposed at least 455 mg/day of vitamin E to produce a significant lipid oxidation-reduction in vivo [77]. Nevertheless, the absence of beneficial activities of individual antioxidants may be explained by the dose-dependent behavior these components exhibit outside their natural matrix, highlighting the critical properties of complex mixtures such as whole foods containing essential micronutrients (vitamins, minerals), dietary fiber, and non-nutrient phytochemicals [47]. Similarly to our survey, a study using the DPPH method investigated the scavenging capacity and synergistic effects of lycopene, vitamin C, vitamin E, and β-carotene [78]. The results showed that the mixture with the highest significant synergistic effects was the solution combining all four compounds [78].

Similar to our study, curcumin’s antioxidant and radical scavenging properties were determined by employing various in vitro antioxidant assays [79], showing that curcumin had an effective DPPH radical scavenging and a high FRAP. They also showed that curcumin effectively inhibited lipid peroxidation of a linoleic acid emulsion [79]. In addition, our study showed that curcumin (similar to coffee) had the highest DPPH and anti-pro-oxidant scores due to its phenolic structure.

Regarding this study’s limitations, as for all models, simulated digestion, as carried out by the INFOGEST consensus model, was not completely physiological. The used enzymes and bile were not of human origin but obtained from porcine sources. Food does not stay in the stomach and the intestine for precisely 2 h; it is variable depending on whether the meal is liquid or solid, the consumed quantity, the density of energy, and how easily digestible it is [80]. Therefore, the 2 h incubation times might not represent digestion accurately, especially considering the various viscosity of the employed matrices (liquid vs. solid). In addition, examining complex food items such as sausage and white chocolate due to the various compounds in their composition and also the variability of the composition of different products can obtain variable results, and we only investigated one food item per food group. However, there are also strengths of this study. The digestion protocol was a standardized consensus model of static digestion [32]. Further strengths include digestions of control samples, and especially measuring both antioxidant and pro-oxidant endpoints of the samples and combining them into a score. Quantities of food items were chosen according to physiologically relevant doses, assuming that our approach (2 g in 25 mL) reflects the proportion of meals in 1 L of gastrointestinal juices (i.e., a 40th of a realistic amount).

## 5. Conclusions

Complex solid matrices (sausage and white chocolate) exhibited both the highest antioxidant and lipid pro-oxidant patterns, while added curcumin and vitamins showed rather the lowest values overall. Liquids generally acted somewhere in the middle range in each test, except for coffee and orange juice, which also had a very high antioxidant potential. Many compounds in these food items, as well as digestive proteins, can influence antioxidant and pro-oxidant measures, and the test values can also depend on the dosages and sensitivities of the chosen tests. These findings should instigate further research and preclude physiological interpretations based on only antioxidant or pro-oxidants alone, as both aspects need to be considered simultaneously. In addition, comparing laboratory combinations with mathematical combinations mainly showed, but not always, additive effects of foods rather than synergistic effects. Thus, scores that integrate several assays of both pro-and antioxidant measures may obtain more comprehensive and accurate results that are more physiologically relevant.

## Figures and Tables

**Figure 1 foods-12-01719-f001:**
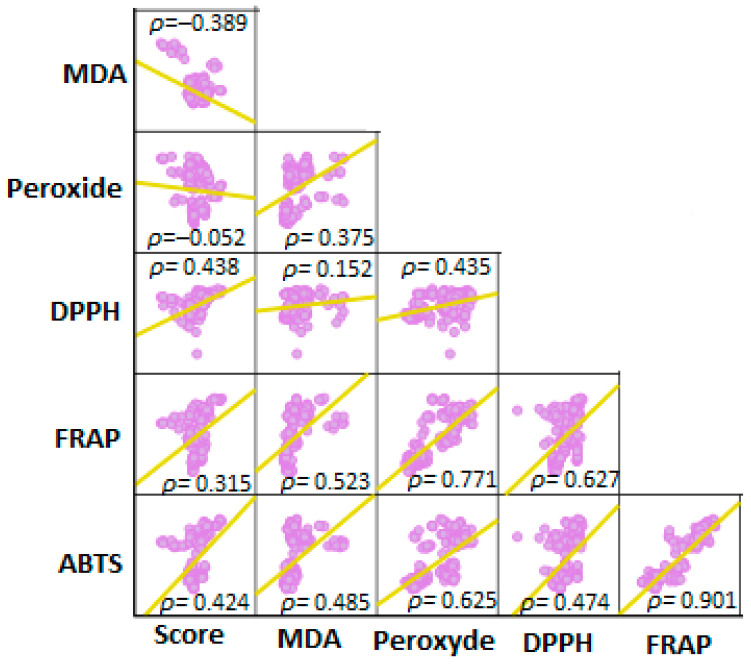
Correlation matrix showing Spearman correlation coefficients (*ρ*) between the pro- and antioxidant measurements and anti-pro-oxidant-score (all *p*-values < 0.001 except for the correlation between peroxides and anti-pro-oxidant score, which was not significant). Highest correlation was between FRAP and ABTS (*ρ =* 0.901). All values are in logarithmic form.

**Table 1 foods-12-01719-t001:** Selected food items, their quantity, and ingredients.

Food Item	Quantity Used per Simulated Digestion	Nutrients (per 100 mL for Liquids and 100 g for Solids)	Ingredients as Listed on Food Labels
**Tomato juice**	2 mL	Fats (<0.5 g), carbohydrates (3 g), starch (0.6 g), proteins (0.8 g), fiber (0.6 g), total energy (16 kcal)	Tomato juice (99.5%), salt (0.5%)
**Orange juice**	2 mL	Fats (0 g), carbohydrates (8.7 g), proteins (0.7 g), vitamin C (12 mg), total energy (42 kcal)	Juice 100%
**Vitamin C**	2.5 mg	-	L-ascorbic acid (99%)
**Vitamin E**	1.5 mg	-	DL-alpha-tocopherol (>97%)
**Coffee**	2 mL	≈8 mg caffeine *, ≈100 mg of total polyphenols **	Coffee, 100% Arabica
**Curcumin**	12.5 mg	-	Curcumin (98%)
**Sausage**	2 g	Fats (29 g), saturated fats (11 g), carbohydrates (7.5 g), proteins (9 g), salt (1.8 g), total energy (329 kcal)	Pork liver 25%, pork liver 22%, pork fat, water, pork rind, potato starch, dextrose, salt, cream, sugar, onions, preservatives (potassium acetate, sodium nitrite), spices, emulsifiers (citric acid esters from mono- and diglycerides from fatty acids), antioxidants (ascorbic acid, sodium ascorbate), thickener (xanthan), spice extracts, hemoglobin, aroma
**Soda**	2 mL	Fats (0 g), carbohydrates/of which sugars (6.6 g/6.6 g), proteins (0 g), total energy (28 kcal)	Sparkling water, sugar, acidifiers: citric acid, malic acid, acidity corrector; sodium gluconate, natural lemon-green lemon aroma, sweetener: steviol glycoside
**White chocolate**	2 g	Fats (25 g), saturated fats (22 g), carbohydrates/of which sugars (55/55 g), proteins (5.7 g), total energy (561 kcal)	Sugar, cocoa butter, whole milk powder, skim milk powder, emulsifier (soya lecithin), flavoring

* https://fdc.nal.usda.gov/fdc-app.html#/food-details/171890/nutrients (accessed on 30 January 2023). ** http://phenol-explorer.eu/contents/food/662 (accessed on 30 January 2023).

## Data Availability

Data is contained within the article or Appendix A.

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
