# Peer review of "Pro- and Antioxidant Effect of Food Items and Matrices during Simulated In Vitro Digestion"

_foods, 2023, doi:10.3390/foods12081719_

Round 1

Reviewer 1 Report

Dear authors,

Thank you for your enormous effort, this study has many improvements required in Foods Journal. The proposed idea needs more improvement, thus providing more novel and scientific soundness. Major comments are attached together with the review result file.

Thank you.

Author Response

Reviewer #1

Dear authors, Thank you for your enormous effort, this study has many improvements required in Foods Journal. The proposed idea needs more improvement, thus providing more novel and scientific soundness. Major comments are attached together with the review result file. Thank you.

Reply:  We thank the reviewer for the time and effort invested in reviewing the study and the feedback; this is much appreciated. We are glad to hear that you find the effort put into the research to be significant. We understand that there are areas of improvement that need to be addressed. We took your comments into account to enhance the scientific soundness of the proposed idea, as outlined in the following.

Abstract: The abstract did not mention / include essential information that should be in it. Please revise the abstract by re-writing the abstract to only contain brief information about 1. Objective of study, 2. Materials and methods, 3. Result trends, 4. Conclusion and study implication.

Reply: We appreciated the reviewer’s comment. We have revised the abstract to cover the essential information better.

Introduction: The language used needs significant improvement. Since, many of the ideas that want to be delivered by authors can not be well understood by readers due to grammatical or language errors.

Reply: We understand the importance of using clear and concise language to communicate ideas to readers effectively. We have revised the language to improve clarity and readability and to structure our ideas better.

Each paragraph within the introduction did not show consistent idea that want to be shown, as an example, paragraph 2 described about the possibility of ROS to worsen or lead to several diseases, and mentioning diabetes and gastrointestinal diseases as an example. It is better to mention the mechanism first of how ROS led to this kind of disease, that mention the GI and an example of it.

Reply: We have revised the introduction to be more straightforward, mentioning more clearly the link between ROS and disease (Lines 42-54, 59-63).

In my objective point of view, I would suggest authors to re-write the introduction to contain all the information necessary and mention some of result trends that utilized food matrices used within the authors’ research.

e.g. :  Paragraph I: Brief explanation of ROS; Paragraph II: Mechanism by which ROS causes diseases and mention example of GI disease; Paragraph III: Effort to reduce the amount of ROS and result trends with regard to the usage of various food matrices; Paragraph IV: Urgencies, strength of this research, wherein authors already understood and well-identified research gap at this field, thus propose the novel approach by performing this research; Paragraph V: Study objectives.

Reply: We appreciate the reviewer’s comment. We have revised the introduction to be more straightforward, following a structure very close to the one recommended.

Materials and Methods: Please mention all details of chemical compounds used, including brand code, company details to contain information of headquarter, city, country.

Reply: We have presented all the information in the main text and further detail in Suppl. Table 1.

L120: The result of those study

Reply:  We agree and have added another reference.

L123: All food items were purchased from a local supermarket and were used as sold?

Reply: We have revised the sentence to be more precise (Line 153). But indeed, all items were used without any further kitchen preparation procedures.

Please suit the format for each table and figure to fit with the format given by the journal author guidelines.

Reply: Following slight revisions, we have used the journal format style.

Table 1 only mentions 9 food items, while authors mention in Line 136 that analysis was conducted on 33 food items or their combination. Why don’t the authors mention all the food items information within the table?

Reply: In fact, we used only 9 food items/nutrients, and the number of combinations used in total, including single food items and their combinations was 33. We have given their composition information in Supplementary Table 1. We have rewritten the relevant section for clarity (Lines 162-168).

It seems like this research just confirmed the anti-oxidant and pro-oxidant potentials of several food matrices by means of gastrointestinal digestion. Therefore, the novelty that this study has is minimum.

Reply: To the best of our knowledge, very limited studies, if any, have systematically investigated the effects of oxidative and/or antioxidant effects of food items and matrices during digestion, using both several pro-and antioxidant endpoints. This is a clear research gap, and we have tried to contribute to obtaining more insights into this domain. In addition, another novelty of our study is the development of the anti-pro-oxidant score, which allowed us to examine the pro- and antioxidant effects comprehensively and together, which will be very useful for future studies. In other words, most studies have rather focused only on either antioxidant effects or pro-oxidant effects but have not investigated both aspects at the same time, which, in our opinion, is an important limitation.

Results and Discussion: The study result highlighted that the blank (without any addition of food matrices) even showed an antioxidant effect by itself, with a significant reducing potential.

Reply: While there is a lack of research on the pro-oxidant effects of bile and digestive enzymes, several studies have found bile components, such as bilirubin, to have an antioxidant effect due to their ability to bind free radicals, e.g., reactive nitrogen species (RNS).

Confusingly, L351 mentioned that the blank showed even higher antioxidant activity than in the FRAP when compared to other food matrices. Vitamin E + soda displayed the lowest antioxidant potential, only showing non-significant differences to the blank. Does it mean that the digestion system would be better without consumption of any of these foods? Please provide more explanation.

Reply: We discussed this finding in the discussion section: While there is a lack of research on the pro-oxidant effects of bile and digestive enzymes, several studies have found bile components, such as bilirubin, to have an antioxidant effect due to their ability to bind free radicals, e.g., reactive nitrogen species (RNS). In addition, the blank does not present higher FRAP antioxidant activity compared to other matrices, it is merely in the same range (Figure 1). The sentence has been revised for clarification.

Reviewer 2 Report

The study is quite interesting, it is well written and discussed, however there are some comments.

Line 229. Why did the authors use AAPH instead of persulfate to generate the ABTS radical?

Line 245. Why measure absorbance at three different wavelengths?

Line 295. Same question. Why measure absorbance at this 520 and 750 nm wavelengths?

The results and discussions are based on the multivariate model, which is fine and complete, however, it would be interesting to have presented the results of each technique with their respective original values, for example in % inhibition, or umol of equivalents per g or mL of sample. , etc. The results presented in logarithms are a bit difficult to understand. I recommend the authors to put these results and discuss about it.

In the discussions, the general part of the structure-activity is missing.

Author Response

Reviewer #2

The study is quite interesting, it is well written and discussed, however there are some comments.

Reply: We thank the reviewer for such promising words; much appreciated.

Line 229. Why did the authors use AAPH instead of persulfate to generate the ABTS radical?

Reply: In general, AAPH and persulfate are both commonly used as radical initiators in the generation of the ABTS radical, which is a widely used chromogenic substrate for the measurement of antioxidant activity. The choice between AAPH and persulfate may depend on several factors, including the specific experimental conditions, the desired reaction kinetics, and the availability of reagents. For example, AAPH may be preferred in certain situations because it generates a more stable radical than persulfate and is less sensitive to pH changes. Additionally, AAPH can be used in a range of solvents, while persulfate may require more specific solvent conditions. Overall, the choice of radical initiator depends on the specific needs and goals of the experiment, and both AAPH and persulfate can be effective choices for generating the ABTS radical.

Line 245. Why measure absorbance at three different wavelengths? Line 295. Same question. Why measure absorbance at this 520 and 750 nm wavelengths?

Reply: In measuring antioxidant activity using the ABTS radical, absorbance is typically measured at 520 nm and 750 nm at two different wavelengths. At 520 nm, the ABTS radical has a maximum absorbance, and this wavelength is used to measure the extent of reduction of the radical by the antioxidant being tested. When an antioxidant is added to the ABTS radical, it donates electrons, reducing its absorbance at 520 nm. Therefore, a decrease in absorbance at 520 nm indicates a higher antioxidant activity of the tested substance. At 750 nm, the absorbance is evaluated to correct for any non-specific absorbance or background noise in the assay, such as turbidity. This wavelength is less affected by the reduction of the ABTS radical and is used to account for any interference that may affect the measurement at 520 nm. For three wavelengths, the same rationale applies – to monitor and correct for background absorption, such as due to turbidity.

The results and discussions are based on the multivariate model, which is fine and complete, however, it would be interesting to have presented the results of each technique with their respective original values, for example in % inhibition, or umol of equivalents per g or mL of sample. , etc. The results presented in logarithms are a bit difficult to understand. I recommend the authors to put these results and discuss about it.

Reply: This is indeed a good point. We understand your concern about presenting the results on logarithmic scales, which can be challenging to understand for some readers. Although the data are not normally distributed and the measurement units of the methods are very different, presenting the results as original data may mislead the article's readers. However, to be more clear in the present manuscript, we now report the minimum and maximum values (ranges) for each measurement on the original scale also.

In the discussions, the general part of the structure-activity is missing.

Reply: We appreciate the reviewer’s comment. We have tried to address this issue briefly in the method section, see Lines 228-233, 261-264, 289-293, 313-322, 346-348 and further in the discussion, see e.g. lines 590-598, 612-620.

Round 2

Reviewer 1 Report

Dear authors,

Thank you for your great effort in revising the manuscript.

I have checked each of the comments from the previous peer-reviewing process and made sure that all comments were answered accordingly.

After a thorough look at the revised manuscript, I stated that this manuscript is acceptable as published article in Foods Journal.

Congratulations.

Reviewer 2 Report

The authors made all the corrections requested. I think it can be publishable.